# Optimization of the Curing and Post-Curing Conditions for the Manufacturing of Partially Bio-Based Epoxy Resins with Improved Toughness

**DOI:** 10.3390/polym11081354

**Published:** 2019-08-15

**Authors:** Diego Lascano, Luis Quiles-Carrillo, Sergio Torres-Giner, Teodomiro Boronat, Nestor Montanes

**Affiliations:** 1Escuela Politécnica Nacional, Ladrón de Guevara E11-253, Quito 17-01-2759, Ecuador; 2Technological Institute of Materials (ITM), Universitat Politècnica de València (UPV), Plaza Ferrándiz y Carbonell 1, 03801 Alcoy, Spain; 3Novel Materials and Nanotechnology Group, Institute of Agrochemistry and Food Technology (IATA), Spanish National Research Council (CSIC), Calle Catedrático Agustín Escardino Benlloch 7, 46980 Paterna, Spain

**Keywords:** bio-based thermosets, post-curing, gel time, mechanical properties

## Abstract

This research deals with the influence of different curing and post-curing temperatures on the mechanical and thermomechanical properties as well as the gel time of an epoxy resin prepared by the reaction of diglycidyl ether of bisphenol A (DGEBA) with an amine hardener and a reactive diluent derived from plants at 31 wt %. The highest performance was obtained for the resins cured at moderate-to-high temperatures, that is, 80 ${}^{{}^{\circ}}\;C$ and 90 ${}^{{}^{\circ}}\;C$, which additionally showed a significant reduction in the gel time. This effect was ascribed to the formation of a stronger polymer network by an extended cross-linking process of the polymer chains during the resin manufacturing. Furthermore, post-curing at either 125 ${}^{{}^{\circ}}\;C\;$or 150 ${}^{{}^{\circ}}\;C$ yielded thermosets with higher mechanical strength and, more interestingly, improved toughness, particularly for the samples previously cured at moderate temperatures. In particular, the partially bio-based epoxy resin cured at 80 ${}^{{}^{\circ}}\;C$ and post-cured at 150 ${}^{{}^{\circ}}\;C$ for 1 h and 30 min, respectively, showed the most balanced performance due to the formation of a more homogeneous cross-linked structure.

## 1. Introduction

Since their early introduction in the 1930s, epoxy resins have been the most used thermosetting polymers available. Epoxy resins show a great range of inherent properties, resulting from their highly reactive epoxy groups located in the terminal chains [1,2]. The resultant outstanding properties make these thermosets suitable for high-performance applications, for instance in aircraft and automotive industries [3,4]. Currently, epoxy systems based on diglycidyl ether of bisphenol A (DGEBA) are widely used in the plastic industry since they present remarkable structural properties including low shrinkage, coating properties, and high chemical resistance [5,6]. Despite this, thermosetting polymer materials are typically known by their intrinsic brittleness due to the high cross-linking density formed during curing [7,8]. Thus, the final properties of a structural epoxy system are not only highly influenced by the type and chemical structure of the monomers and the curing agent, which is the cross-linking precursor, but also by the curing conditions and external factors, such as the curing temperature, pressure, and so forth [9,10].

A potential advantage of epoxy systems is their capability to tailor their final properties on the basis of the selection of the processing conditions and parameters. In this context, choosing the type of cross-linking agent is one of the most effective methods. Different amine hardeners, anhydride hardeners, and acid hardeners have been tested to develop epoxy-type thermosetting polymers [11,12,13,14]. Among them, amine hardeners are the most employed as they can provide cross-linking at relatively low temperatures [15,16]. The stoichiometric ratio between the epoxy resin components and the hardener should be carefully chosen. Any variation on this ratio may cause an excess or lack of either epoxide or amine groups, modifying, in a direct way, the epoxy system properties [17]. This effect is particularly relevant for the glass transition temperature (T_g_) of the resultant epoxy resins [18,19]. In this regard, many research studies have found that the optimal properties are achieved at the stoichiometric point [17,20]. Another strategy to modify their properties is based on the addition of some diluents or “flexibilizers” [21,22]. Besides changing viscosity and toughness, these additives are also added for manufacturing purposes, since epoxy resins may present an excessively short pot-life that can lead to an earlier gelation process, making the systems difficult to handle [23].

In recent years, a wide range of several renewable building blocks and green composites [24] have been pursued in the polymer industry, with the aim to reduce the dependence on petrochemical feedstocks. Industry can take advantage of the long chains of vegetable oils, which are made of three fatty acids, that is, triglycerides structures linked to a glycerol base molecule [25]. Vegetable oils present interesting features due to their carbon–carbon double bonds and the high reactivity provided by the multiple functional groups [26]. The addition of vegetable oils to conventional resins can successfully reduce the fossil content of synthetic resins and, thus, partially decouple them to petroleum. Then, several research works have been recently devoted to explore substitutes obtained from renewable resources to produce epoxy systems. For instance, Jaillet et al. [27] successfully synthesized a polyacid hardener from soybean of higher reactivation, which could potentially replace amine hardeners. In another study, Stemmelen et al. [28] synthesized aminated fatty acid (AFA) hardeners for curing epoxidized linseed oil (ELO), observing a beneficial effect on the thermochemical properties due to the improved oxidation profile of the epoxy resin.

The improvement of the final properties of a given epoxy system can also be obtained by optimizing the curing process [29]. In particular, the application of low temperatures during curing can yield a thermosetting resin with a low T_g_, since some reactive groups, either from the epoxy resins or hardeners, do not react completely [30]. Thereafter, post-curing is habitually needed in order to maximize the final T_g_ value [31]. Post-curing is habitually set at a higher temperature than that applied during curing in order to reach the optimal state of cross-linking of the thermoset [32,33]. This process results in a resin with improved mechanical properties that, in combination to the low shrinkage attained, offers the highest stability. Furthermore, it can also provide greater resistance in coating surfaces and higher adhesion strength, making the resultant resins suitable for several industrial applications such as paints, adhesives, high-performance membranes, and so on [34,35,36].

The objective of this work is to study and optimize the temperature applied during curing and post-curing of a partially bio-based epoxy resin. The newly produced thermoset was evaluated in terms of its mechanical, morphological, rheological, and thermomechanical properties.

## 2. Experimental

### 2.1. Materials

The commercial partially bio-based epoxy system was composed of an epoxy resin and an amine-based type hardener supplied as Resoltech^®^ 1070 ECO and Resoltech^®^ 1074 ECO, respectively, by Castro Composites (Pontevedra, Spain). The epoxy resin is particularly made of a mixture of petroleum-derived DGEBA and a plant-based reactive diluent from vegetable oil epoxidation. As indicated by the supplier, the bio-based content of the cured resin is 31 wt %, as determined by ASTM D6866-12. The chemical structure of the main reactive components of the epoxy resin and the cross-linking/hardening system are shown in Figure 1.

### 2.2. Resin Manufacturing

The bio-based epoxy resin and the hardener were added under the stoichiometric ratio 100:35 (*wt/wt*) in a glass vessel, following the manufacturer recommendations. To this end, the bio-based epoxy resin and the curing agent were thoroughly mixed manually for 5 min and then casted into a silicon mold. The resultant mixture has a density of 1.22 g·cm^−3^ and viscosity of 700 mPa·s and also, according to the manufacturer, its T_g_ can reach values of up to 73 ${}^{{}^{\circ}}\;C$. Further details about the uncured mixture can be found in our previous research [37]. All samples were cured for 1 h and subsequently post-cured for 30 min. The different curing and post-curing temperatures are summarized in Table 1.

### 2.3. Mechanical Tests

Flexural tests were performed according to ISO 178 on rectangular pieces with dimensions of 4  × 10 ×  80 mm^3^ in a mechanical universal testing machine ELIB 50 from S.A.E. Ibertest (Madrid, Spain) with a 5 kN load cell. The cross-head speed was set at 10 mm·min^−1^. The impact strength was determined by the Charpy test, following the recommendations of ISO 179, in a Charpy pendulum from Metrotec (San Sebastián, Spain) using a 1-J pendulum. Hardness was obtained following ISO 868 in a durometer 676-D from J. Bot Instruments (Barcelona, Spain). Tests were carried out at room temperature using, at least, six specimens of each sample.

### 2.4. Microscopy

The morphology of the fracture surfaces of the bio-based epoxy resins obtained from the impact tests was analyzed by using field-emission scanning electron microscopy (FESEM) in a ZEISS ULTRA 55 FESEM microscope from Oxford Instruments (Abingdon, UK). An acceleration voltage of 2 kV was used. All samples’ surfaces were previously coated with an ultrathin gold–palladium layer in a high-vacuum sputter coater EM MED20 from Leica Microsystem (Milton Keynes, UK).

### 2.5. Thermomechanical and Rheological Tests

The thermomechanical properties of the bio-based epoxy resins were evaluated by dynamic mechanical thermal analysis (DMTA) in an oscillatory rheometer AR-G2 from TA Instruments (New Castle, DE, USA). Tests were carried out in torsion–shear conditions, with a special clamp system made for solid samples. The samples, sizing 4 × 10 × 40 mm^3^, were subjected to a heating program from 30 ${}^{{}^{\circ}}\;C$ to 140 ${}^{{}^{\circ}}\;C$ at a heating rate of 2 ${}^{{}^{\circ}}\;C$·min^−1^. The curing behavior was studied by means of a system of parallel plates of 25 mm diameter made for liquid samples, with a constant gap of 0.2 mm. These samples were extended over the preheated plates and subjected to an isothermal heating program at 70 ${}^{{}^{\circ}}\;C$, 80 ${}^{{}^{\circ}}\;C,$ and 90 ${}^{{}^{\circ}}\;C$ for 1 h. In both cases, a maximum deformation (γ) of 0.1% and a constant frequency of 1 Hz were set. The gel time was obtained from the crossover point between the storage modulus (*G*′) and the loss modulus (*G*″), in *G*′ = *G*″ conditions, according to Equation (1):
(1)$$t_{gel}=C\;\frac{1}{k}$$
where $t_{gel}$ is the gel time, *C* is a constant, and *k* is the reaction rate constant that can be parameterized through the Arrhenius expression, shown in Equation (2):(2)$$k=A\;e^{-\frac{E_{a}}{R\;T}}$$
where *A* is the frequency factor, *E_a_* is the apparent activation energy, *R* refers to the universal gas constant, and *T* is the absolute temperature. By substituting Equation (2) in Equation (1), the following Equation (3) was obtained:(3)$$t_{gel}=C^{\prime }\;\frac{1}{e^{-\frac{E_{a}}{R\;T}}}$$

Then, Equation (4) resulted by taking natural logarithms on both sides of Equation (3):(4)$$\ln(t_{gel})=C^{\prime \prime }+\frac{E_{a}}{R\;T}$$

Finally, by plotting $\ln(t_{gel})$ versus $\frac{1}{T}$, the value of *E_a_* was calculated through the slope of the linear fitting.

## 3. Results and Discussion

### 3.1. Mechanical Properties of Partially Bio-Based Epoxy Resins

The flexural properties of the partially bio-based epoxy resins after curing and post-curing are gathered in Table 2. The value of the flexural modulus (*E_f_*) of the bio-based epoxy resin cured at 70 ${}^{{}^{\circ}}\;C$ was 977 ± 127 MPa, whereas its flexural strength ($\sigma_{f}$) was 77.4 ± 13.4 MPa, thus both values were relatively high. One can observe that the curing temperature increase led to an increase of the flexural strength properties of the thermosetting system. In particular, *E_f_* increased to values of 1260 ± 192 MPa and 2403 ± 210 MPa when the samples were cured at 80 ${}^{{}^{\circ}}\;C\;$and 90 ${}^{{}^{\circ}}\;C$, respectively, representing an increase of 191% and 245% when compared with the specimen cured at 70 ${}^{{}^{\circ}}\;C$. This effect can be ascribed to the fact that the application of high curing temperatures promotes cross-linking of the thermosetting resin and then yields a stronger polymer network, in which the density of the cross-linked chains increases and the chain mobility is reduced [38]. In relation to post-curing, all the samples showed a further mechanical strength improvement by the application of the thermal post-treatment at both 125 ${}^{{}^{\circ}}\;C\;$and 150 ${}^{{}^{\circ}}\;C$. However, one notices that the highest improvement in the flexural properties was attained by subjecting the samples to, first, a curing at 80 ${}^{{}^{\circ}}\;C$ and, subsequently, a post-curing at 150 ${}^{{}^{\circ}}\;C$. This bio-based epoxy resin reached *E_f_* and $\sigma_{f}$ values of 3237 ± 377 MPa and 123.9 ± 7.6 MPa, respectively, which represents an increase of approximately 231% and 60.1% compared with the specimen cured at 70 ${}^{{}^{\circ}}\;C$ without any thermal post-treatment. Unexpectedly, the partially bio-based epoxy resins cured at 90 ${}^{{}^{\circ}}\;C$ and then post-cured at either 125 ${}^{{}^{\circ}}\;C$ or 150 ${}^{{}^{\circ}}\;C$ showed lower mechanical values than those obtained for the samples previously cured at 80 ${}^{{}^{\circ}}\;C$ and subjected to the same post-curing conditions. The decrease observed in the mechanical resistance for the sample cured at the highest temperature can be ascribed to an excessive increase of the cross-linking degree that potentially yields a polymer structure with a heterogeneous load distribution and also a wide distribution of gaps between the cross-linked regions. In this regard, Gupta et al. [31] and also Bueche [39] reported that, although the mechanical performance of epoxy-type thermosetting resins habitually tends to increase with the cross-linking degree, excessively cross-linked structures can also promote a decrease of the average number of cross-linking points. Therefore, the use of excessive high temperatures can potentially restrict the formation of homogeneously cross-linked structures by limiting the molecular diffusion during the resin manufacturing. The resultant phenomenon increases the chance that polymer chains achieve a tension large enough to break, thus reaching a final value of stress at break below the one initially expected to occur.

The hardness at the Shore D scale and the impact-absorbed energy values, obtained by the Charpy test, are also gathered in Table 2. In relation to hardness, the values followed the same tendency as observed above for *E_f_* and $\sigma_{f}$ since this property is related to the mechanical resistance properties. Similarly, the Shore D hardness value increased from 77.4 ± 2.9, for the partially bio-based epoxy resin cured at 70 ${}^{{}^{\circ}}\;C$ without any thermal post-treatment, up to 85.3 ± 0.5, for the specimens cured at 80 ${}^{{}^{\circ}}\;C$ and post-cured at 125 ${}^{{}^{\circ}}\;C$ or 150 ${}^{{}^{\circ}}\;C.$ In the case of the impact strength, it can be observed that all the partially bio-based epoxy resins showed no dependency on the curing temperature, that is, the samples absorbed nearly the same quantity of energy, of about 12 kJ·m^−2^, after being cured at 70 ${}^{{}^{\circ}}\;C$, 80 ${}^{{}^{\circ}}\;C$, and 90 ${}^{{}^{\circ}}\;C$. Interestingly, the partially bio-based epoxy resins developed improved toughness when subjected to the different post-curing treatments. Thus, the samples cured at 70 ${}^{{}^{\circ}}\;C$, 80 ${}^{{}^{\circ}}\;C$, and 90 ${}^{{}^{\circ}}\;C$ and thereafter post-cured at 125 ${}^{{}^{\circ}}\;C$ showed values of 12.2 ± 1.9 kJ·m^−2^, 13.3 ± 1.7 kJ·m^−2^, and 15.9 ± 3.7 kJ·m^−2^, respectively. The maximum energy absorption was also observed for the resin cured at 80 ${}^{{}^{\circ}}\;C$ and post-cured at 150 ${}^{{}^{\circ}}\;C$, which showed an impact-strength value 16.8 ± 2.5 kJ·m^−2^. Although fracture resistance tends to decrease when the cross-linking density increases [40], this property can be additionally dependent on other structural parameters. For instance, Min et al. [41] reported that toughness is also affected by the free volume, chain flexibility, and degree of the intermolecular packing of the thermoset. Therefore, it can increase at temperatures below T_g_. This effect is explained by the fact that the conformation configuration of the individual bonds is modified and, thus, the resultant free volume varies due to the rotation of the benzene rings in the epoxy resin.

### 3.2. Morphology and Density of Partially Bio-Based Epoxy Resins

The FESEM images of the surface fractures from the impact tests are shown in Figure 2. Left and right images gather the FESEM micrographs taken at high and low magnification, respectively. The images corresponding to the specimen cured at 80 ${}^{{}^{\circ}}\;C$ without being post-cured are presented in Figure 2a,b. As it can be seen, the surface fractures were uniform, showing many crack growths. The surfaces are typical of a brittle fracture caused by the absence of plastic deformation [42], which is in accordance to the mechanical properties reported above. One can observe a morphological change in the fracture surface of the partially bio-based epoxy resin that was subjected to post-curing at 125 ${}^{{}^{\circ}}\;C$, shown in Figure 2c,d. These images revealed the existence of a ductile fracture, presenting some rough and tearing zones. A similar morphology can be observed in the sample post-cured at 150 ${}^{{}^{\circ}}\;C$, included in Figure 2e,f. Alternatively, one can observe in Figure 2g,h that the surface fracture corresponding to the sample subjected to curing and post-curing at 70 ${}^{{}^{\circ}}\;C$ and 125 ${}^{{}^{\circ}}\;C$, respectively, was relatively smooth and uniform. This surface is typical of a brittle fracture, and it was relatively similar to that observed for the sample cured at 80 ${}^{{}^{\circ}}\;C$ without any post-curing treatment. In this regard, Min et al. [41] indicated that epoxy systems can unexpectedly present a more ductile fracture behavior at higher curing temperatures. This observation is in agreement with the above-reported mechanical properties, since both samples showed similar values of impact strength. A rougher surface can be observed in Figure 2i,j, which correspond to the partially bio-based epoxy resin subjected to curing at 90 ${}^{{}^{\circ}}\;C$ and post-curing at 125 ${}^{{}^{\circ}}\;C.$ The morphology attained indicates that the fracture surface was relatively ductile, as supported by the mechanical analysis, due to the fact that both the use of high curing temperatures and the application of a post-curing treatment leads to an increase in mechanical ductility.

The densities of the partially bio-based epoxy resins after the different curing and post-curing processes were also evaluated in order to ascertain the effect of the thermal treatments on the resin free volume. The values of density are summarized in Table 3. One notices that the density values of the samples significantly increased from 1.15 ± 0.02 g·cm^−3^, for a curing temperature of 70 ${}^{{}^{\circ}}\;C$, to 1.60 ± 0.26 g·cm^−3^, for 80 ${}^{{}^{\circ}}\;C$. This effect is in accordance with the literature, where it is proposed that by increasing the curing temperature, the cross-linking density increases, thus obtaining stiffer materials [38]. However, the sample cured at 90 ${}^{{}^{\circ}}\;C$ showed a density of only 1.19 ± 0.01 g·cm^−3^, which can be ascribed to the formation of the aforementioned excessive cross-linked structure with a higher free volume. In this regard, Gupta and Brahatheeswaran [43] proposed that, although increasing the curing temperature generally yields cross-linking density increases, an opposite effect can be attained at excessive high temperatures, in which high degrees of cross-linking results in an increase in the resin free volume due to the formation of an heterogeneously cross-linked structure. Therefore, as described above, the use of excessive high temperatures can not only increase the curing speed but also limits the molecular diffusion process during the resin formation, resulting in cross-linked structures with lower mechanical performance and also lower densities [39]. Moreover, one can observe that the post-curing treatment also induced a noticeable effect on the density of all the partially bio-based thermosetting resins. It can be observed that, in all cases, the specimens subjected to post-curing at 125 ${}^{{}^{\circ}}\;C$ showed an unexpected decrease of density, which are in agreement to the increase of the sample free volume due to a low intermolecular packing of the thermoset. Interestingly, a more extended cross-linking between the resin molecules and the hardener molecules was attained in the specimens that were cured at 70 ${}^{{}^{\circ}}\;C\;$and 80 ${}^{{}^{\circ}}\;C$ and subsequently post-cured at 150 ${}^{{}^{\circ}}\;C$, since their density presented an increase of 21% and 9%, respectively, compared with the samples thermally post-treated at 125 ${}^{{}^{\circ}}\;C$. Otherwise, the samples cured at 90 ${}^{{}^{\circ}}\;C$ and post-cured at 150 ${}^{{}^{\circ}}\;C$ presented a lower density value. Therefore, for samples cured at moderate temperatures, that is, 70 ${}^{{}^{\circ}}\;C\;$and 80 ${}^{{}^{\circ}}\;C$, the use of high temperatures during post-curing can result advantageous to obtain thermosets with a cross-linked structure that is more homogenous. This observation suggests that, on the basis of the toughness improvement reported above, in addition to cross-linking, other phenomena affecting the intermolecular packing occurred in the thermoset.

### 3.3. Rheological Properties of Partially Bio-Based Epoxy Resins

The curing processes of the partially bio-based epoxy resin carried out at 70 ${}^{{}^{\circ}}\;C$, 80 ${}^{{}^{\circ}}\;C$, and 90 ${}^{{}^{\circ}}\;C$ were studied by oscillatory rheometry, and the results are gathered in Figure 3. In the evolution of *G*′ with time, shown in Figure 3a, low values can be observed at the early stages of the curing process. However, as cross-linking occurred, the M_W_ of the bio-based epoxy resin increased and, thus, the values of *G*′ increased, improving the material elastic behavior. Figure 3b shows the phase angle (δ) between the obtained deformation (γ) and the applied stress (σ). At the first step of the curing process, the δ value was close to 90°, which is characteristic of a liquid behavior. The curing process was identified by the decrease of δ with time, being reduced to approximately 0°. Therefore, for an elastic solid material, this rheological behavior confirms that the curing process fully ended.

The gel time is one of the most important parameters from a manufacturing point of view since it delimits the change from the liquid to the solid state and it is noticed by an increase in the mixture viscosity. The gel point of these polymer networks occurred in a conversion around 0.7. This parameter can be estimated either when δ is 45° or by the crossover point between *G*′ and *G*″. Table 4 summarizes the gel time values at different curing temperatures. As mentioned above, the cross-linking process ended when the δ value was 0°. One can observe that the gel time decreased from 1426.1 s, at the curing temperature of 70 ${}^{{}^{\circ}}\;C,$ to 445.2 s, when the curing temperature was set at 90 ${}^{{}^{\circ}}\;C$. Then, as expected, the gel time decreased markedly with the curing temperature increase, since high temperatures can significantly speed up the cross-linking process. This observation represents a valuable indicator of the cross-linking reaction course. In particular, for a curing temperature of 70 ${}^{{}^{\circ}}\;C$, the curing time was 2250.3 s, whereas it decreased significantly to 750.2 s when the curing temperature increased to 90 ${}^{{}^{\circ}}\;C$. Alternatively, the maximum storage modulus (*G*′_max_) values were achieved when the samples were completely cured. These values are also gathered in Table 4, in which one can observe that the highest value was attained for the sample cured at 80 ${}^{{}^{\circ}}\;C$, which is in accordance with the mechanical properties reported above.

*E_a_* was obtained through the slope of the linear fitting of the plot of $\ln(t_{gel})$ versus $\frac{1}{T}$, shown in Figure 4. It resulted in a value of 60.28 kJ·mol^−1^, having an excellent correlation coefficient ($R^{2}\approx 0.999)$. This is in agreement with our previous work dealing with the kinetic analysis of the curing process of DGEBA-based epoxy resins using bio-based diluents, in which the *E_a_* values ranged between 50 kJ·mol^−1^ and 70 kJ·mol^−1^ [37]. The *E_a_* value of the here-prepared partially bio-based epoxy resin is similar, for instance, to the values reported for conventional epoxy–amine systems, that is, 58–59 kJ·mol^−1^ [44], but higher than those of epoxy–anhydride systems, that is, 46–48 kJ·mol^−1^ [45]. In this regard, Karger-Kocsis et al. [46] indicated that *E_a_* increased from 53 kJ·mol^−1^ to 72 kJ·mol^−1^ by increasing the vegetable oil content in the epoxy system.

### 3.4. Thermomechanical Properties of Partially Bio-Based Epoxy Resins

Figure 5 shows the thermomechanical behavior of the bio-based epoxy resins after the different curing and post-curing treatments. Table 5 summarizes the values of *G*′ at 40 ${}^{{}^{\circ}}\;C$ and 110 ${}^{{}^{\circ}}\;C$ and also of T_g,_ obtained from the DMTA curves. Figure 5a–c depict the evolution of *G*′ with temperature for the samples that were subjected to curing at 70 ${}^{{}^{\circ}}\;C$, 80 ${}^{{}^{\circ}}\;C,$ 90 ${}^{{}^{\circ}}\;C$ and post-curing at 125 ${}^{{}^{\circ}}\;C$ and 150 ${}^{{}^{\circ}}\;C$. At 40 ${}^{{}^{\circ}}\;C$, the values of *G*′ were 1.029 ± 0.021 GPa, 1.039 ± 0.025 GPa, and 1.037 ± 0.024 GPa for the resins cured at 70 ${}^{{}^{\circ}}\;C$, 80 ${}^{{}^{\circ}}\;C,$ and 90 ${}^{{}^{\circ}}\;C,$ respectively. As the temperature increased, a sharp drop in the *G*′ values of two orders of magnitude was observed, indicating that the partially bio-based epoxy resins underwent glass transition. At a higher temperature, that is, 110 ${}^{{}^{\circ}}\;C$, the samples showed similar *G*′ values, that is, 4.29 ± 0.09 MPa, 4.99 ± 0.15 MPa, and 4.68 ± 0.09 MPa, for each respective curing temperature. It is also worthy to mention that higher *G*′ values were attained with the curing temperature increase. This result can be explained by the fact that the increase in the cross-linking density results in a material’s stiffness [47].

Furthermore, the effect of the post-curing treatment was noticeable, especially for the partially bio-based epoxy resins that were thermally post-treated at 150 ${}^{{}^{\circ}}\;C.$ These samples showed a significant improvement in elasticity, reaching values of 1.255 ± 0.027 GPa, 1.048 ± 0.022 GPa, and 1.221 ± 0.02 GPa at a temperature of 40 ${}^{{}^{\circ}}\;C$, and values of 8.15 ± 0.16 GPa, 6.96 ± 0.17 GPa, and 7.47 ± 0.14 GPa at 110 ${}^{{}^{\circ}}\;C$, for curing temperatures of 70 ${}^{{}^{\circ}}\;C$, 80 ${}^{{}^{\circ}}\;C$, and 90 ${}^{{}^{\circ}}\;C$, respectively. Therefore, the partially bio-based epoxy resin cured and post-cured at 70 ${}^{{}^{\circ}}\;C$ and 150 ${}^{{}^{\circ}}\;C$, respectively, showed the highest *G*′ values. However, it is worth emphasizing that, despite the fact that the post-curing treatment tended to increase *G*′, the samples cured at 80 ${}^{{}^{\circ}}\;C$ and 90 ${}^{{}^{\circ}}\;C$ and then subjected to post-curing at 150 ${}^{{}^{\circ}}\;C$ presented lower *G*′ values at 40 ${}^{{}^{\circ}}\;C$ than those post-cured at 125 ${}^{{}^{\circ}}\;C$. In particular, unexpected low *G*′ values were attained for the resins cured at 80 ${}^{{}^{\circ}}\;C\;$and then post-cured at either 125$\;{}^{{}^{\circ}}\;C$ or 150 ${}^{{}^{\circ}}\;C$, even though they presented a highly cross-linked structure. In this regard, Dyakonov et al. [48] indicated that elasticity in epoxy resins is not only related to an increase in the cross-linking density but also to a phenomenon of resin vitrification. Therefore, it is proposed that resin vitrification was successfully prevented for the samples cured at moderate temperatures and then post-cured at high temperature, that is, 150 ${}^{{}^{\circ}}\;C$, due to a more efficient molecular diffusion. This result supports the improved toughness of the partially bio-based epoxy resin observed during the mechanical analysis, and it offers a new technological advantage from a mechanical point of view.

The evolution of the damping factor (*tan δ*) with temperature is shown in Figure 5d–f. The *tan δ* peak relates to the T_g_ of the biopolymer. The resins cured at 70 ${}^{{}^{\circ}}\;C$, 80 ${}^{{}^{\circ}}\;C$, 90 ${}^{{}^{\circ}}\;C$ presented T_g_ values of approximately 62.5 ± 1.25 ${}^{{}^{\circ}}\;C$, 65.1 ± 1.63 ${}^{{}^{\circ}}\;C$, 70.4 ± 1.47 ${}^{{}^{\circ}}\;C$, respectively. It is noteworthy that by increasing the curing temperature, the cross-linking degree was extended, and it resulted in an increase of T_g_ [49]. The effect of post-curing on the thermomechanical behavior of the samples was noticeable since it favored the functional groups to react after the curing process. Higher temperatures promoted the cross-linking process to continue and led to a further increase in the T_g_ values. For instance, the partially bio-based epoxy resins that were subjected to curing at 70 ${}^{{}^{\circ}}\;C$ or 80 ${}^{{}^{\circ}}\;C$ and subsequent post-curing at 125 ${}^{{}^{\circ}}\;C$ had an increase of 28% and 45%, respectively, in the T_g_ values when compared with their counterpart resin without post-curing. In particular, the latter samples reached values of 90.5 ± 1.71 ${}^{{}^{\circ}}\;C$ and 84.9 ± 1.95 ${}^{{}^{\circ}}\;C$. In the case of the partially bio-based epoxy resins subjected to post-curing at 150 °C, the increase in T_g_ was even more significant. In particular, the values obtained after post-curing at 150 ${}^{{}^{\circ}}\;C$ were 94.1 ± 1.97 °C, 93.4 ± 1.87 ${}^{{}^{\circ}}\;C,$ and 95.3 ± 3.01 ${}^{{}^{\circ}}\;C$ for the samples previously cured at 70 °C, 80 ${}^{{}^{\circ}}\;C,$ and 90 ${}^{{}^{\circ}}\;C$, respectively. Wu [50] suggested that T_g_ stabilization relates to a restriction of the mobility degree of the polymer chains. Therefore, the T_g_ values attained correlate with the previous density measurements, confirming the reduction of the sample free volume due to the successful extension of the cross-linking process by the formation of a more homogeneous structure. In addition, the partially bio-based epoxy resins subjected to post-curing at 125 ${}^{{}^{\circ}}\;C$ presented lower T_g_ values than those post-cured at 150 ${}^{{}^{\circ}}\;C$, even though the latter samples showed higher toughness.

## 4. Conclusions

The effect of the curing and post-curing conditions on epoxy resins prepared by the reaction of DGEBA with amine hardener and a bio-based reactive diluent at 31 wt % was analyzed by flexural tests, FESEM, and oscillatory rheometry at different isothermal heating rates. It was observed that the partially bio-based thermosetting resins cured and post-cured at higher temperatures tended to present higher mechanical and thermomechanical performance due to the formation of a highly cross-linked structure of lower free volume. However, interestingly, the samples cured at moderate temperatures and then post-cured at high temperatures unexpectedly developed improved toughness performance. In particular, it was observed that curing at 80 ${}^{{}^{\circ}}\;C$ for 1 h and subsequent post-curing at 150 ${}^{{}^{\circ}}\;C$ for 30 min yielded the partially bio-based epoxy resin with the most balanced mechanical and thermomechanical properties. This result was ascribed to the formation of a more homogeneous cross-linked structure based on a strong polymer network but with also improved chain mobility. Based on the above, it can be concluded that the final properties of these epoxy resin systems can be effectively controlled by the selection of the curing and post-curing temperatures. The application of the optimal conditions can then yield partially bio-based epoxy resins with improved toughness that represent an environmentally friendly solution for the thermosetting industry. Moreover, they can be manufactured in a shorter production time since the gel time is also reduced.

## Figures and Tables

**Figure 1 polymers-11-01354-f001:**
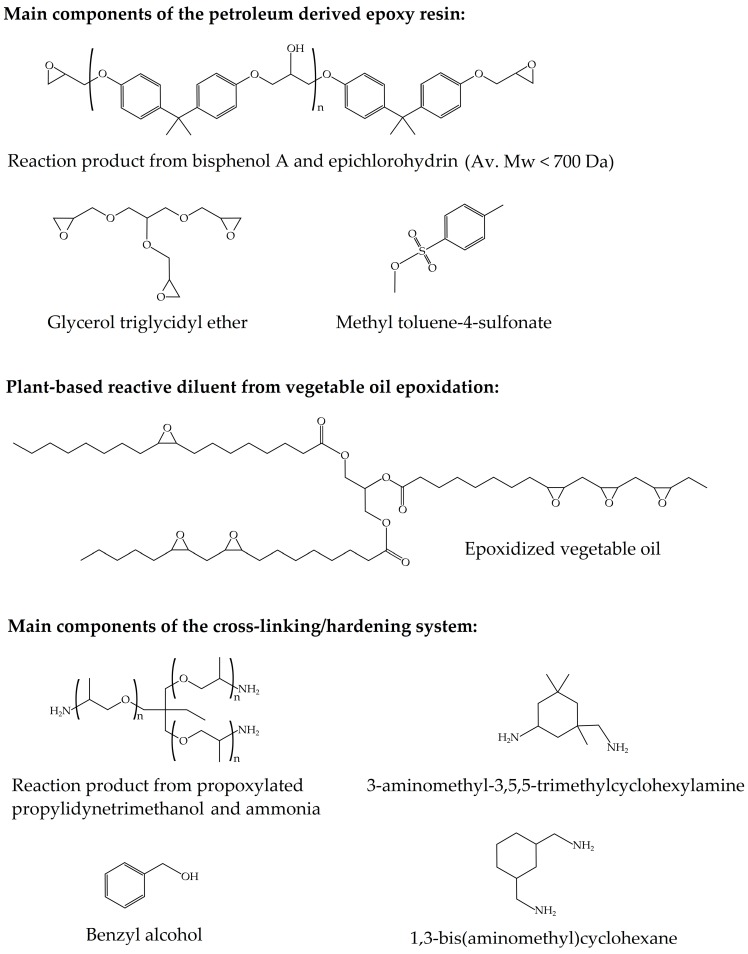
Main reactive components of the partially bio-based epoxy system.

**Figure 2 polymers-11-01354-f002:**
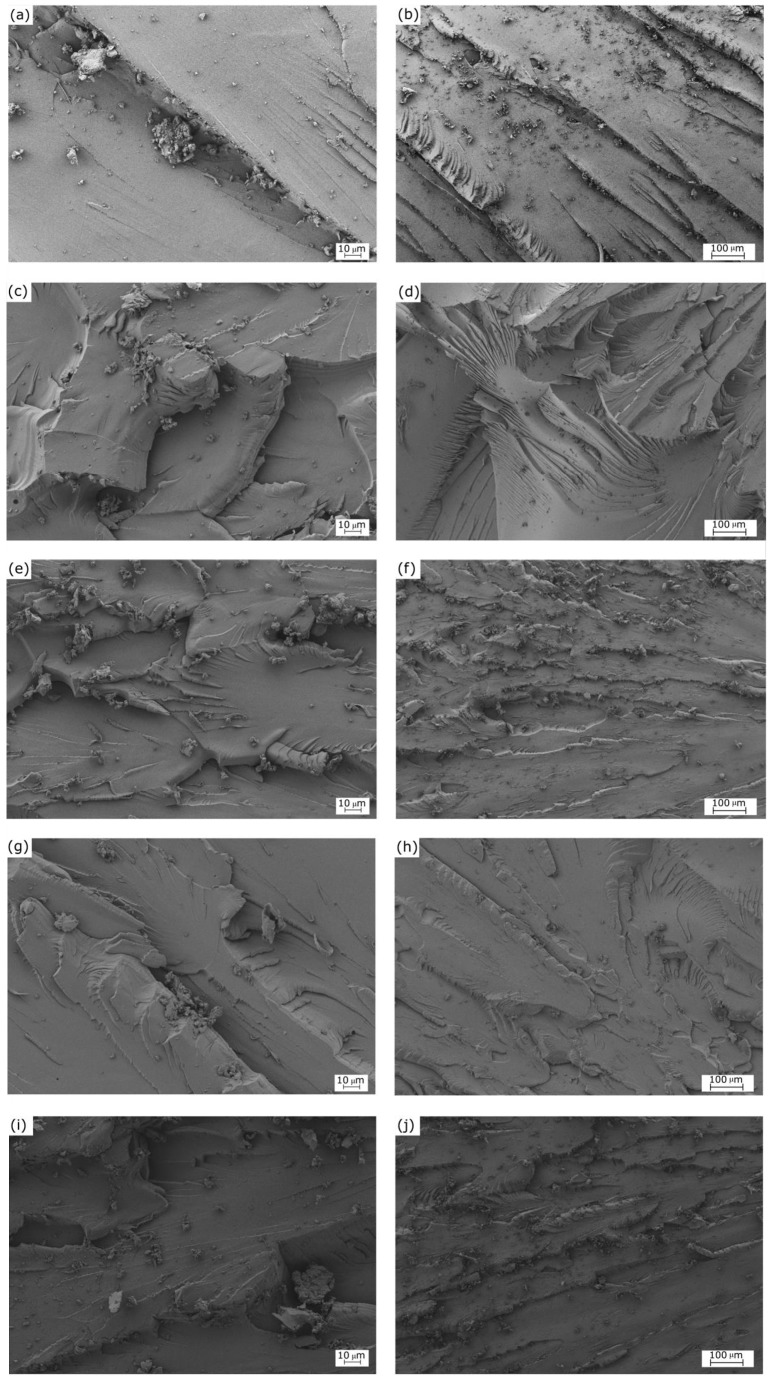
Field-emission scanning electron microscopy (FESEM) corresponding to the fracture surfaces from impact tests of the partially bio-based epoxy resins subjected to standard curing (SC) and post-curing (PC) treatments: (**a**,**b**) SC_80_; (**c**,**d**) SC_80_PC_125_; (**e**,**f**) SC_80_PC_150_; (**g**,**h**) SC_70_PC_125;_ (**i**,**j**) SC_90_PC_125_. Images were taken at 500× (left column) and 100× (right column), with scales of 10 µm and 100 µm, respectively.

**Figure 3 polymers-11-01354-f003:**
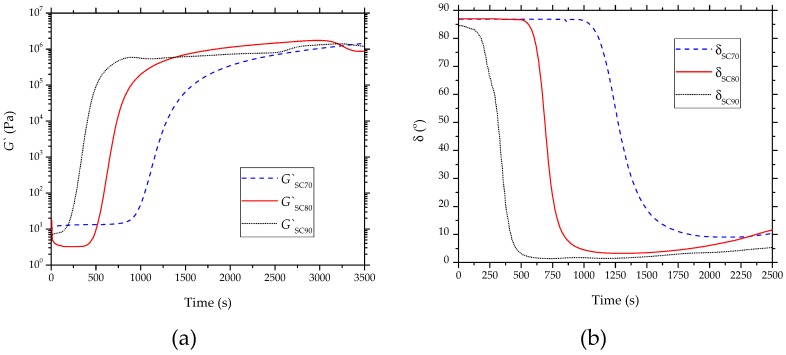
Evolution of (**a**) storage modulus (*G*′) and (**b**) phase angle (δ) with time of the partially bio-based epoxy resins subjected to standard curing (SC) treatment, obtained by rheometry.

**Figure 4 polymers-11-01354-f004:**
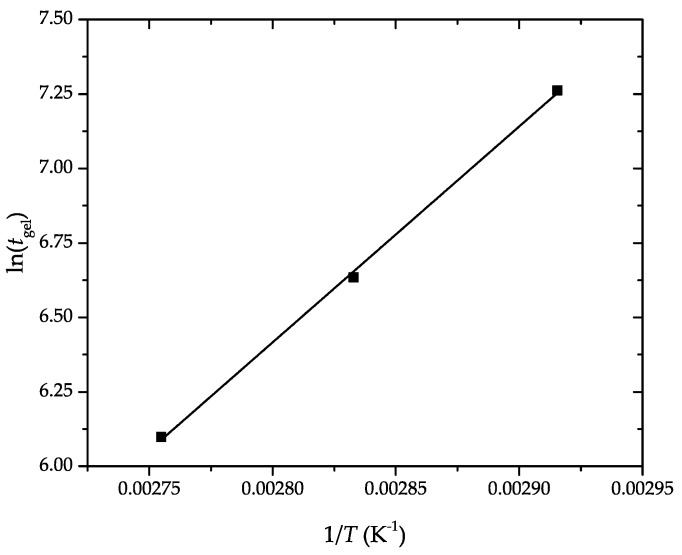
Determination of the apparent activation energy (*E_a_*) of the partially bio-based epoxy resins by the linear fitting of the gel time versus the inverse temperature, according to Equation (4).

**Figure 5 polymers-11-01354-f005:**
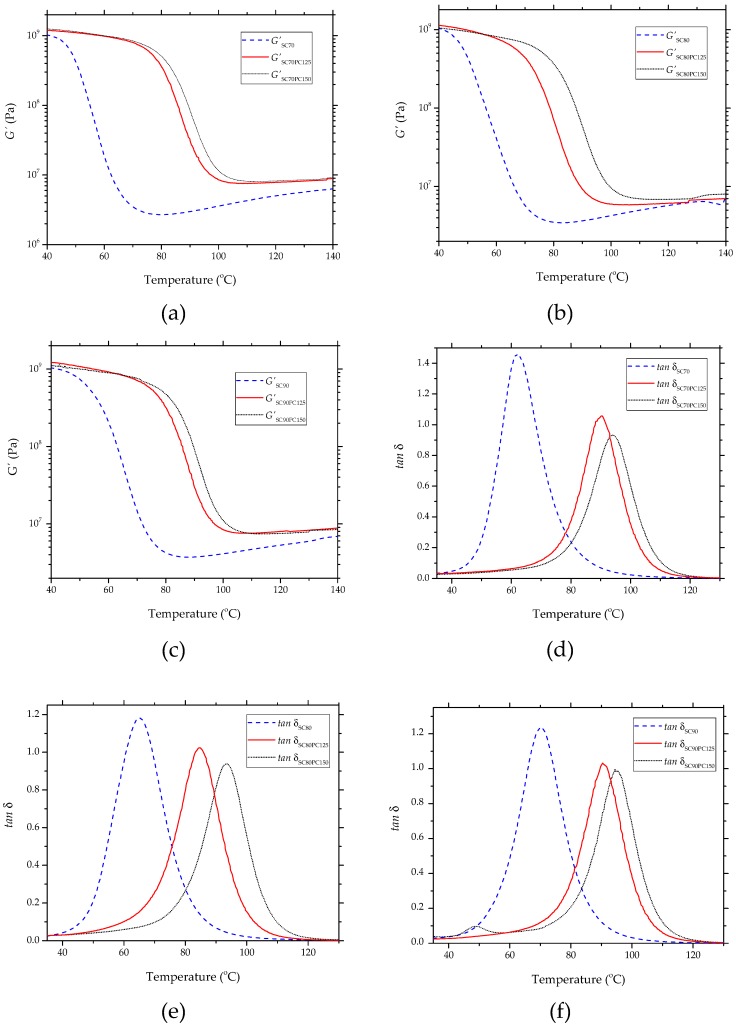
Thermomechanical behavior of the partially bio-based epoxy resins subjected to standard curing (SC) and post-curing (PC) treatments: (**a**–**c**) storage modulus (*G*′); (**d**–**f**) damping factor (*tan δ).*

**Table 1 polymers-11-01354-t001:** Composition and labelling of the partially bio-based epoxy resins according to the selected curing and post-curing temperatures. The standard curing (SC) and the post-curing (PC) treatments were applied for 1 h and 30 min, respectively.

Resin	$T_{\mathrm{curing}}\;({}^{{}^{\circ}}\;C)$	$T_{\mathrm{post}-\mathrm{curing}\;}({}^{{}^{\circ}}\;C)$
SC_70_	70	-
SC_70_PC_125_		125
SC_70_PC_150_		150
SC_80_	80	-
SC_80_PC_125_		125
SC_80_PC_150_		150
SC_90_	90	-
SC_90_PC_125_		125
SC_90_PC_150_		150

**Table 2 polymers-11-01354-t002:** Flexural modulus (E*_f_*), and flexural strength ($\sigma_{f}$), Shore D hardness, and impact strength of the partially bio-based epoxy resins subjected to standard curing (SC) and post-curing (PC) treatments.

Resin	Flexural Test		Shore D Hardness	Impact Strength (kJ·m^−2^)
	E_f_ (MPa)	$\sigma_{f}\;(\mathrm{MPa})$		
SC_70_	977 ± 127	77.4 ± 13.4	77.4 ± 2.9	11.9 ± 2.1
SC_70_PC_125_	1766 ± 391	89.2 ± 13.1	81.3 ± 1.2	12.2 ± 1.9
SC_70_PC_150_	1854 ± 256	93.2 ± 22.2	80.0 ± 1.2	12.8 ± 1.3
SC_80_	1260 ± 192	81.1 ± 15.9	82.2 ± 1.8	11.5 ± 1.6
SC_80_PC_125_	2379 ± 185	114.4 ± 22.0	85.3 ± 1.0	13.3 ± 1.7
SC_80_PC_150_	3237 ± 377	123.9 ± 7.6	85.3 ± 0.5	16.8 ± 2.5
SC_90_	2403 ± 210	105.6 ± 10.3	85.0 ± 1.5	12.0 ± 0.9
SC_90_PC_125_	2520 ± 298	110.6 ± 6.3	84.3 ± 0.6	15.9 ± 3.7
SC_90_PC_150_	2207 ± 164	101.1 ± 12.3	83.4 ± 1.7	12.4 ± 1.2

**Table 3 polymers-11-01354-t003:** Density of the partially bio-based epoxy resins subjected to standard curing (SC) and post-curing (PC) treatments.

Resin	Density (g·cm^−3^)
SC_70_	1.15 ± 0.02
SC_70_PC_125_	1.11 ± 0.05
SC_70_PC_150_	1.35 ± 0.17
SC_80_	1.60 ± 0.26
SC_80_PC_125_	1.32 ± 0.07
SC_80_PC_150_	1.45 ± 0.19
SC_90_	1.19 ± 0.01
SC_90_PC_125_	1.07 ± 0.04
SC_90_PC_150_	0.97 ± 0.17

**Table 4 polymers-11-01354-t004:** Gel time (t_gel_), curing time (t_curing_), and maximum storage modulus (*G*′_max_) of the partially bio-based epoxy resins after the different curing temperatures.

$T{\mathrm{curing}}_{\;}({}^{{}^{\circ}}\;C)$	t_gel_ (s)	t_curing_ (s)	*G*′_max_ (GPa)
70	1426.1 ± 28.5	2250.3 ± 56.1	1.042 ± 0.02
80	760.6 ± 15.2	1250.0 ± 18.8	1.738 ± 0.03
90	445.2 ± 8.9	750.2 ± 18.7	1.311 ± 0.03

**Table 5 polymers-11-01354-t005:** Storage modulus (*G*′) at 40 ${}^{{}^{\circ}}\;C$ and 110 ${}^{{}^{\circ}}\;C$ and glass transition temperature (T_g_) of the partially bio-based epoxy resins subjected to standard curing (SC) and post-curing (PC) treatments.

Resin	$G^{\prime }\;\mathrm{at}\;40\;{}^{{}^{\circ}}\;C$ (GPa)	$G^{\prime }\;\mathrm{at}\;110\;{}^{{}^{\circ}}\;C$ (MPa)	$T_{g}\;({}^{{}^{\circ}}\;C)$
SC_70_	1.029 ± 0.021	4.29 ± 0.09	62.5 ± 1.25
SC_70_PC_125_	1.203 ± 0.024	7.56 ± 0.15	90.5 ± 1.71
SC_70_PC_150_	1.255 ± 0.027	8.15 ± 0.16	94.1 ± 1.97
SC_80_	1.039 ± 0.025	4.99 ± 0.09	65.1 ± 1.63
SC_80_PC_125_	1.132 ± 0.019	5.89 ± 0.11	84.9 ± 1.95
SC_80_PC_150_	1.048± 0.022	6.96 ± 0.17	93.4 ± 1.87
SC_90_	1.037 ± 0.024	4.68 ± 0.09	70.4 ± 1.47
SC_90_PC_125_	1.230 ± 0.025	7.57 ± 0.15	90.7 ± 2.35
SC_90_PC_150_	1.121 ± 0.02	7.47 ± 0.14	95.3 ± 3.01

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
