# Peer review of "Optimization of the Curing and Post-Curing Conditions for the Manufacturing of Partially Bio-Based Epoxy Resins with Improved Toughness"

_polymers, 2019, doi:10.3390/polym11081354_

Round 1

Reviewer 1 Report

Epoxy resins have found their wide applications in many fields. The addition of plant-based epoxy reactive diluents into epoxy resins can reduce the dependence on petrochemical feedstocks and improve the properties of resins. The characteristics of epoxy resins can be tailored by changing precursors, hardeners, and diluents as well as the processing conditions and parameters. This manuscript discusses the effects of curing and post-curing temperatures on the mechanical and thermomechanical properties as well as the gel time of partially bio-based epoxy resins. The research accords with the scope of Polymers. The paper is well organized, the research results are good, the important viewpoints are clearly presented, and the conclusions are supported by the complete data. The paper can be accepted for publishing.

Please note the following minor issues:

Line 159: “yielded” should be replaced by “yields”.

Line 210: “led to” might be replaced by “leads to”.

Lines 238-239: I suggest authors to improve the following sentence: “ --- the use of high temperatures during post-curing can result advantageous to obtain thermosets with ---” (result advantageous to).

Line 271: “in” (the highest value was attained in the sample) should be replaced by “for”.

Line 292: The replacement of “and then post-cured” by “and post-curing” would be better.

Lines 299-300: “the cross-linking density increased, resulting in a material’s stiffness” might be replaced by “the increase in the cross-linking density results in a material’s stiffness” or “the increased cross-linking density results in a material’s stiffness”.

Lines 313: “despite the post-curing treatment tended to increase G’” should be replaced by “despite the fact that the post-curing treatment tended to increase G”.

Author Response

Epoxy resins have found their wide applications in many fields. The addition of plant-based epoxy reactive diluents into epoxy resins can reduce the dependence on petrochemical feedstocks and improve the properties of resins. The characteristics of epoxy resins can be tailored by changing precursors, hardeners, and diluents as well as the processing conditions and parameters. This manuscript discusses the effects of curing and post-curing temperatures on the mechanical and thermomechanical properties as well as the gel time of partially bio-based epoxy resins. The research accords with the scope of Polymers. The paper is well organized, the research results are good, the important viewpoints are clearly presented, and the conclusions are supported by the complete data. The paper can be accepted for publishing.

Please note the following minor issues:

Line 159: “yielded” should be replaced by “yields”.

This word was corrected.

Line 210: “led to” might be replaced by “leads to”.

This word was changed as indicated by the reviewer.

Lines 238-239: I suggest authors to improve the following sentence: “ --- the use of high temperatures during post-curing can result advantageous to obtain thermosets with ---” (result advantageous to).

This sentence was modified as indicated by the reviewer.

Line 271: “in” (the highest value was attained in the sample) should be replaced by “for”.

This word was replaced.

Line 292: The replacement of “and then post-cured” by “and post-curing” would be better.

This change was also done.

Lines 299-300: “the cross-linking density increased, resulting in a material’s stiffness” might be replaced by “the increase in the cross-linking density results in a material’s stiffness” or “the increased cross-linking density results in a material’s stiffness”.

This sentence was also modified as indicated by the reviewer.

Lines 313: “despite the post-curing treatment tended to increase G’” should be replaced by “despite the fact that the post-curing treatment tended to increase G”.

As suggested by the reviewer, this expression was modified.

Reviewer 2 Report

Can the authors tell readers the structure of theplant-based epoxy reactive diluent? It is quite important to understand the curing process. The specimens cured at 90 ℃ exhibited inferior mechanical resistances and small density, which was ascribed to an excessive increase of the cross-linking degree, it seems to be difficult to understand. Higher temperature usually induces quick curing speed , but not higherdegrees of cross-linking. The authors should give more convincing interpretation.

Author Response

Can the authors tell readers the structure of the plant-based epoxy reactive diluent? It is quite important to understand the curing process. The specimens cured at 90 ℃ exhibited inferior mechanical resistances and small density, which was ascribed to an excessive increase of the cross-linking degree, it seems to be difficult to understand. Higher temperature usually induces quick curing speed, but not higher degrees of cross-linking. The authors should give more convincing interpretation.

As suggested by the reviewer, the composition of the epoxy system was better described in section 2.1 and the chemical structure of all its components was  included in a new Figure 1. As indicated by the reviewer, we have also improved some sentences in the manuscript to better explain the effect of the curing temperature on the partially bio-based resins. Briefly, the use of high temperatures during curing favors the extension of the cross-linking process but it also restricts the formation of homogeneously cross-linked structures by limiting the molecular diffusion. Thus, excessive high temperatures during the resin manufacturing can potentially result in resin vitrification, generating materials with lower mechanical performance and also lower densities. For this reason, the use of moderate temperatures during curing and the application of post-curing treatments at higher temperatures results advantageous. Please see the more detailed explanation added in page 5 lines 168-170, page 8 lines 233-239 or page 12 lines 330-332.